# Sensitivity of Storm Response to Antecedent Topography in the XBeach Model

**Rangley C. Mickey [1],\* , Patricia S. Dalyander [2] , Robert McCall [3] and Davina L. Passeri [1]**

1   U.S. Geological Survey St. Petersburg Coastal and Marine Science Center; St. Petersburg, FL 33701, USA; dpasseri@usgs.gov
2   The Water Institute of the Gulf, Baton Rouge, LA 70802, USA; sdalyander@thewaterinstitute.org
3   Deltares, 2629 Delft, The Netherlands; Robert.mccall@deltares.nl
\*   Correspondence: rmickey@usgs.gov

**Abstract:** Antecedent topography is an important aspect of coastal morphology when studying and forecasting coastal change hazards. The uncertainty in morphologic response of storm-impact models and their use in short-term hazard forecasting and decadal forecasting is important to account for when considering a coupled model framework. This study provided a methodology to investigate uncertainty of profile response within the storm impact model XBeach related to varying antecedent topographies. A parameterized island Gaussian fit (PIGF) model generated an idealized baseline profile and a suite of idealized profiles that vary specific characteristics based on collated observed LiDAR data from Dauphin Island, AL, USA. Six synthetic storm scenarios were simulated on each of the idealized profiles with XBeach in both 1- and 2-dimensional setups and analyzed to determine the morphological response and uncertainty related to the varied antecedent topographies. Profile morphologic response tends to scale with storm magnitude but among the varied profiles there is greater uncertainty in profile response to the medium range storm scenarios than to the low and high magnitude storm scenarios. XBeach can be highly sensitive to morphologic thresholds, both antecedent and time-varying, especially with regards to beach slope.

**Keywords:** numerical modeling; XBeach; morphodynamic; dune; beach; synthetic storm; antecedent topography; uncertainty

## 1. Introduction

On barrier coasts, the morphodynamic response of barrier islands to major storms is critical in the assessment of coastal resiliency and vulnerability [1]. While storm characteristics are inherently important, antecedent morphology has been shown to be a primary factor controlling beach response to extreme events [2,3]. A variety of models have been developed to aid coastal impact assessment [2,4–7]. The degree of sensitivity of each model to the antecedent morphology (e.g., fore-dune height and width, beach slope and width, etc.) varies depending on model complexity and use. Specifically, simple parameterized models, such as the Sallenger storm impact scale [2], and the Stockdon et al., hurricane response model [8], are sensitive to morphologic thresholds related to antecedent topography since they rely on present or historic data for model inputs of beach slope, as well as dune crest and dune base elevations for impact classification. More complex morphodynamic process-based models, such as XBeach [6], can also be highly sensitive to the initial morphologic conditions [9–11] and simulation results may vary depending on specific characteristics of the pre-storm morphology.

Due to this dependence of modeled beach response to antecedent conditions, efforts to model the response of a barrier island or mainland beach to a storm event would ideally incorporate accurate

pre-storm morphology rather than approximate or outdated morphology. The prohibitive cost of acquiring elevation data through LiDAR (Light Detecting and Ranging) or other methods immediately preceding a storm event typically means that there is considerable uncertainty in the antecedent morphology. This uncertainty tends to be largest for the beach and nearshore regions, which can evolve rapidly over timescales ranging from days to weeks and under low magnitude conditions. Although dunes are more stable features that recover on timescales of years following higher magnitude storm events [12,13], antecedent dune morphology for a storm may still have uncertainty associated with accretion over time or erosion during storm events that occur following the most recent elevation survey. There can also be systematic errors or offsets in elevation data that do not result from real-world change, which need to be accounted for and corrected before the data can be used in any analytical method [14].

Recently, there has been a push towards the use of storm impact models to operationally predict coastal hazards [10,15–17], and the use of storm impact models in coupled modeling systems to predict long-term barrier island response [18–21]. Both applications enhance the need to understand the sensitivities of the storm-impact models to antecedent conditions to evaluate how this uncertainty influences the accuracy of predicted storm response, and ultimately the implications for a given application. For example, if the primary objective of the coupled model system is to estimate the vulnerability of a restoration project to storms over a 10-year project lifetime, a recovery model only needs to resolve the pre-storm profile to the level of accuracy that influences the response predicted by the storm-impact model. Additionally, the same concept could be applied to existing but out-of-date morphologic data for regions forecasted to be impacted by storms within an operational framework. Quantifying how uncertainty in initial model elevations propagates through storm impact models will enhance understanding of the predictive skill of model frameworks applied over storm- or decadal-time scales.

In the current study, a methodology was developed to systematically assess the sensitivity and uncertainty of morphologic change resulting from the storm-impact model XBeach [6]. XBeach is a process-based numerical storm-impact model that solves for wind/swell waves, infragravity waves, flow, sediment transport and morphological change, and has been well-validated for storm impacts on barrier islands [22–25]. Evolution of idealized barrier island profiles with varying characteristics of the antecedent topography (beach width, beach slope, fore-dune width and height, fore-dune cross shore position, island base height, and the presence of a berm feature) were simulated under varying hydrodynamic conditions to identify what their effect is on the overall predicted morphological response.

An outline of the paper is as follows. Section 2 (Materials and Methods) describes the methods related to (i) extracting real-world morphologic profile characteristics from LiDAR data; (ii) generation of idealized baseline and idealized modified profiles; (iii) description of simulated storm scenarios and predicted storm impact regime classification; and (iv) numerical model setup. Section 3 (Results) presents the morphological response for all profile scenarios and uncertainty derived from comparison between modified and baseline profiles. Section 4 (Discussion) discusses the morphologic impacts associated with storm magnitude and the identification of uncertainties related to antecedent topography and morphologic response. Section 5 (Conclusions and Future Work) discusses the main conclusions along with future work and expansion of the proposed methodology.

## 2. Materials and Methods

### 2.1. Study Area and LiDAR Analysis

This methodology was tested at Dauphin Island, AL, USA, a low-lying barrier island off the coast of Alabama in the northern Gulf of Mexico (Figure 1). Dauphin Island has been impacted by and recovered from several major storms over the past two decades; the storms have historically overwashed and inundated portions of the island [26]. The elevation data from Dauphin Island

used in this study provides a range of profile characteristics that describe various island states (pre-storm, post-storm, recovered). Eleven LiDAR datasets were analyzed to determine the variability of the island dune and beach characteristics. This study focused on island features from the western portion of Dauphin Island (west of the post-Hurricane Katrina breach of 2005, commonly referred to as Katrina Cut; Figure 1), which is uninhabited and not as anthropogenically influenced as the area east of Katrina Cut [27]. Dauphin Island represents a composite barrier island formed from Pleistocene barrier ridges on the eastern portion and Holocene sediments comprising the narrow western portion [28]. This western portion of Holocene sediments is highly influenced by the dominant westward littoral drift and is extremely vulnerable to island overwash and breaching during storms [29]. This vulnerability leads to major variations in island morphology throughout the time span of the provided LiDAR dataset due to storm impacts which provide a range of antecedent morphologic conditions for this study.

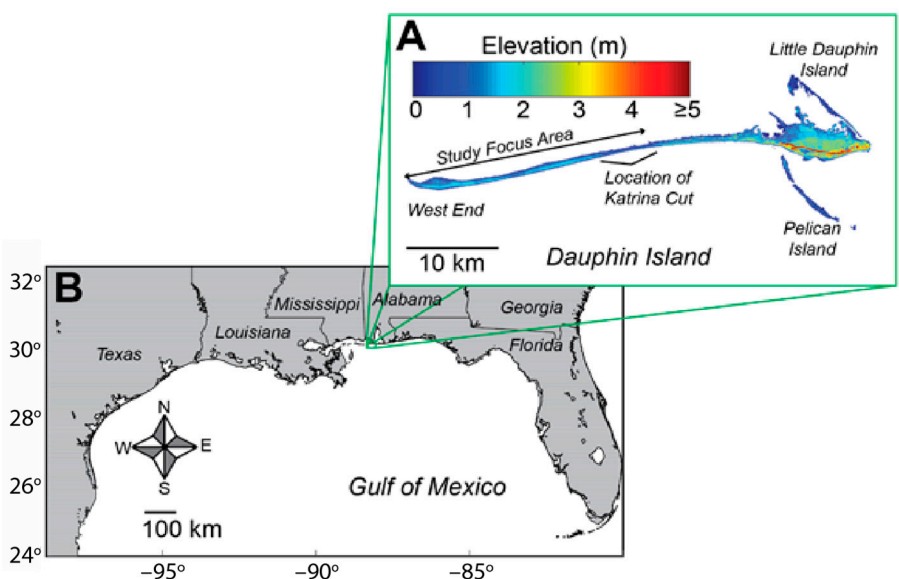

**Figure 1.** Location of Dauphin Island (**A**) within the northern Gulf of Mexico (**B**) (modified from [21]).

The LiDAR datasets span approximately 10 years from September 2005 to January 2015 [30]. These datasets cover different morphological stages of island configurations, including a post-Hurricane Katrina survey that characterized a degraded island and a recovered configuration of island features represented by the latest survey collected in 2015 [31]. Features already extracted from the LiDAR dataset [30], include fore-dune position, fore-dune height (relative to NAVD 88), fore-dune toe (fore-dune base derived from the location of maximum slope change from fore-dune crest to shoreline identified using methods of [32], and shoreline position taken as the mean high water (MHW) level position at 0.23 m (m; NAVD88) obtained from the National Oceanic and Atmospheric Association's (NOAA) National Data Buoy Center (NDBC) tide gauge at Dauphin Island (8735180). Shore-normal profiles were extracted from gridded LiDAR data with an alongshore resolution of 10 m and a cross-shore resolution of 2.5 m [33]. In some cases, fore-dune height, fore-dune toe, or shoreline position were not identifiable in LiDAR-derived profiles. For example, if measurements were taken at high tide, the MHW line may be obscured; these profiles were excluded from the analyses. Beach width was calculated as the horizontal distance from the fore-dune toe to the shoreline (as illustrated in Figure 2A), and beach slope was calculated using an endpoint method following methods from [33].

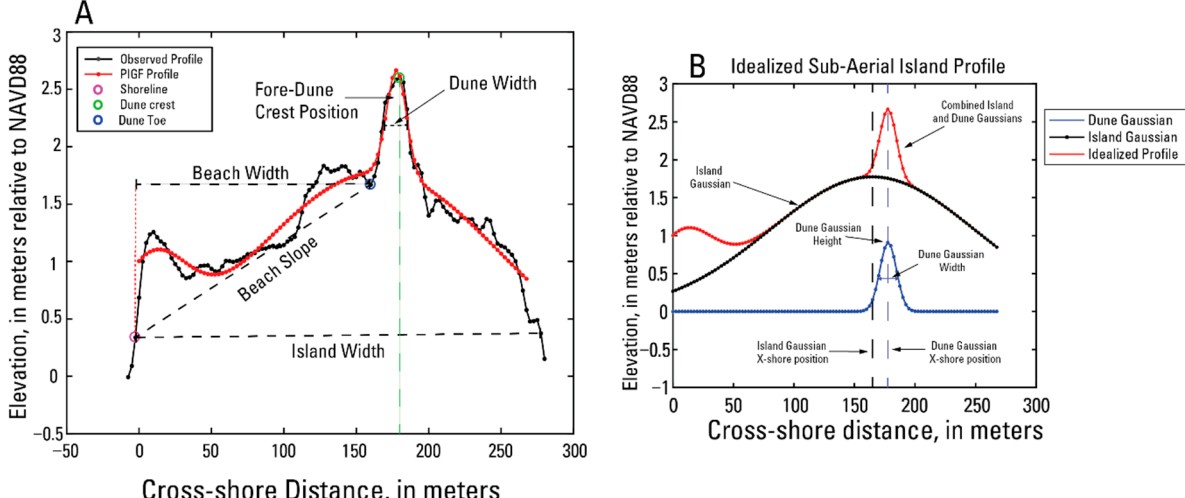

**Figure 2.** (**A**) Example of parameterized island gaussian fit (PIGF) profile (red) fitted to an observed profile (black) from the June 2008 LIDAR survey, with location indicators where profile features were extracted or calculated. (**B**) Sum of Gaussians with labeled individual parts that make up the final idealized profile in (**A**).

While many characteristics were derived directly from the LiDAR profile, others were obtained by applying a parameterized island Gaussian fit (PIGF) model, developed by [30], which provided a systematic estimation of other characteristics that were not defined previously. The advantage of the PIGF model is it can be applied to large datasets to quickly provide well defined features that robustly characterize a cross-shore profile. The PIGF model fits Gaussian curves to specific features of the observed profiles to generate a smoothed island profile consisting of the sum of multiple Gaussian curves with the profile beginning at the 0 m elevation (NAVD88), designated the 0 m cross-shore position, and terminating at various lengths landward of that depending on the input variable thresholds (set to 0 m elevation). One Gaussian curve represents the sub-aerial island centroid, and one or more curves represent dunes and berms, if present (Figure 2B). Applying the PIGF model provides an estimate of relative fore-dune height above the island centroid Gaussian curve at that dune position compared to the fore-dune height typically extracted that is measured as an absolute elevation above relative datum. Having this relative dune height is important since the absolute height would vary across the profile depending on where the island centroid Gaussian is positioned versus the dune feature position. The PIGF model also provides a more consistent estimation of fore-dune width, calculated as the full width at half maximum of dune Gaussian height versus the dune width based on position of dune toe. Fore-dune and island centroid Gaussian positions were measured relative to the 0 m elevation (NAVD88). Occasionally, the best fit of combined Gaussian curves identified with the PIGF model did not properly identify the island centroid and dune features (for example, the solution amalgamated multiple features into one Gaussian rather than resolving the individual features), see [21] Appendix 2 for model details. These profiles were excluded from analysis by specifying a 100 m maximum threshold for fore-dune widths. Elevation features were adjusted to correct for systematic offsets identified in [14] for these LiDAR surveys.

After island features were extracted, they were averaged and interpolated to 200 m in the alongshore through application of a 400 m wide Hanning window, following methods of [33]. The mean, 5th, and 95th percentile of characteristics over all profiles and times were then calculated to establish the range of values for sensitivity testing (Table 1). Since berms were not a persistent feature in time or space, characteristic values for berm width and height were derived by averaging the Gaussian derived berm heights and widths of selected profiles from the 2015 LiDAR survey wherein berm features were identified (Table 1). A doubling of the mean berm height, to 0.5 m, was established arbitrarily as an endmember "high" value for berm crest elevation.

**Table 1.** Dauphin Island beach, dune, and island feature height and width means, highs (95th percentile), and lows (5th percentile), in meters (elevations relative to NAVD88; slope in m/m).

| Characteristic | Mean | High | Low |
|---|---|---|---|
| | | 95th % | 5th % |
| Beach Width | 61.03 | 73.33 | 33.68 |
| Beach Slope | 0.022 | 0.028 | 0.017 |
| Fore-Dune Height | 2.06 | 2.61 | 1.24 |
| Fore-Dune Width | 33.93 | 49.41 | 27.11 |
| Fore-Dune Gauss Height | 0.95 | 1.39 | 0.40 |
| Fore-Dune Gauss cross-shore Position | 153.29 | 170.57 | 75.97 |
| Berm Height | 0.25 | 0.50 [1] | - |
| Berm Width | 15.8 | - | - |
| Island Gauss Height | 1.33 | 1.74 | 0.94 |
| Island Gauss Width | 274.11 | - | - |
| Island Gauss cross-shore Position | 115.41 | - | - |

[1] High berm height was arbitrarily set as double the mean value.

## 2.2. Generation of Baseline and Varied Idealized Profiles

The baseline characteristic profile was generated by applying the mean values for each characteristic listed in Table 1 to a modified version of original PIGF model that includes an additional function for a linear beach portion. This linear beach portion is joined to form a piecewise continuous island profile including the sum-of-Gaussians approximation for the combined island base centroid and fore-dune curves.

$$
\begin{aligned}
z_{island}(x) \;=\; & \left\{ (x > x_T) \times G_{high,cent} \times e^{\left[-\left(\frac{x - G_{x,cent}}{0.6005612 \; \times G_{width,cent}}\right)\right]^2} \right\} \\
& + \left\{ (x \le x_T) \times \left[ mx + \left( G_{high,cent} \times e^{\left[-\left(\frac{x_T - G_{x,cent}}{0.6005612 \; \times G_{width,cent}}\right)\right]^2} - m \times x_T \right) \right] \right\}
\end{aligned}
\tag{1}
$$

Equation (1) uses the cross-shore position of the beach and dune transition point (represented by $x_T$; derived from shoreline position plus the mean beach width), along with the beach slope ($m$), island Gaussian cross-shore position ($G_{x,cent}$), height ($G_{high,cent}$), and width ($G_{width,cent}$) to generate the base profile of the island ($z_{island}$) for the length of the cross-shore coordinates ($x$) (Figure 2B black dotted line). A Gaussian curve representing the fore-dune was derived from mean parameters (fore-dune cross-shore position, $G_{x,dune}$; height, $G_{high,dune}$; and width, $G_{width,dune}$; Figure 2B blue line) and added to the $z_{island}$ profile to generate the full baseline profile in Figure 2B (red line).

A characteristic offshore elevation profile was generated using a pre-Hurricane Ivan (2004) digital elevation model (DEM) developed by [24]. Offshore bathymetry data from a relatively alongshore uniform section of western Dauphin Island (Figure 3A; red box) was averaged alongshore to provide a mean profile of offshore bathymetry, which was connected to the idealized island profile generated from the modified PIGF model. The averaged offshore profile extended from the 0.23 m MHW line to a water depth of approximately 16 m. To preserve the calculated mean beach width of the idealized island profile portion, the offshore profile portion was adjusted positively or negatively in elevations ≤0.8 m depending on the difference in the shoreward end point elevation of each idealized island profile (Figure 3B); this adjustment provided for a seamless connection to each of the idealized island profiles without adjusting island profile elevations. The cross-shore resolution of the profiles is variable and ranges from approximately 12.5 m offshore to 2.5 m from the shoreline to the back barrier. The baseline and modified profiles represent idealized states of the island to inform how XBeach evolves varying antecedent morphologic conditions.

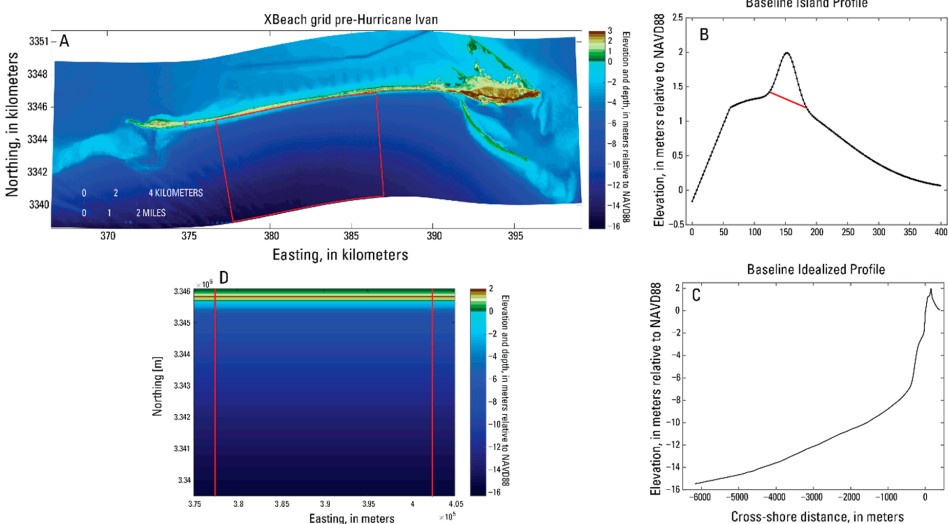

**Figure 3.** (**A**) Digital elevation model (DEM) of Dauphin Island from 2004, prior to Hurricane Ivan [24]. Red box indicates area used to calculate an alongshore average offshore depth profile to use in the current study. (**B**) Baseline island idealized profile generated using mean values for island, dune, and beach profile characteristics; red line indicates dune footprint area examined in post-storm analyses for dune volume and width changes. (**C**) Seamless topographic and bathymetric idealized profile generating by combining the baseline idealized island profile and alongshore averaged bathymetric profile. (**D**) Alongshore uniform 2D grid generated by replication of the idealized profile in (**C**). Area east and west of the vertical red lines were excluded from analysis because of wave shadowing effects.

The sensitivity of model results to varying initial conditions was tested by individually modifying parameters of the baseline profile by replacement of the mean value with high (95th) and low (5th) percentile values of profile characteristics. To add a berm to the baseline profile, another set of Gaussian values (mean berm height, width, and position) was added to the modified PIGF model. Sensitivity testing was also conducted to evaluate the response of the model to varying heights of the island Gaussian curve, while keeping the absolute height (fore-dune height and island height combined) of the profile fixed. The afore-mentioned modifications resulted in a total of 15 different island profiles: one baseline profile (Figures 3B and 4E); eight profiles based on the 95th and 5th percentile values of beach width, beach slope, fore-dune height, and fore-dune width (Table 1, Figure 4A,B,F,I,J,M,N); two profiles with a berm present seaward of the baseline fore-dune (Figure 4C,D); two profiles with varying island centroid heights but consistent overall island heights (dune plus island centroid; Figure 4K,L), and two profiles with modified dune positions that vary seaward and landward of the baseline position (Figure 4G,H). The modified profile with the low value of fore-dune height was excluded from analysis because the maximum elevation is lower than the elevation of the island centroid at the cross-shore location of the placed fore-dune. Evaluation of observed profiles indicates that this inconsistency results from correlation in island characteristics, specifically that the lowest fore-dune heights correspond to post-storm profiles where the island centroid is also in a deflated state. In general, it should be noted that individual characteristics of realistic profiles will tend to be cross-correlated and defined by environmental forcing (e.g., seasonal steepening and narrowing of the beach, erosion of the island centroid and fore-dune, etc.). For the current study, these parameters were individually modified to allow for sensitivity testing of the model to specific antecedent characteristics. Additionally, the complexity in applying fixed values for inherently correlated features must be noted, with specific regards to varying beach slope while trying to maintain a fixed beach width and vice versa. The modified beach width profiles (Figure 4A,B) illustrate this model complexity through the linear beach portion of the profile generated by the modified PIGF model. This linear portion is wider for the high beach width profile (Figure 4B) compared to the low beach width profile (Figure 4A) while the overall beach width, as described in Figure 2A from shoreline to dune toe, is narrower.

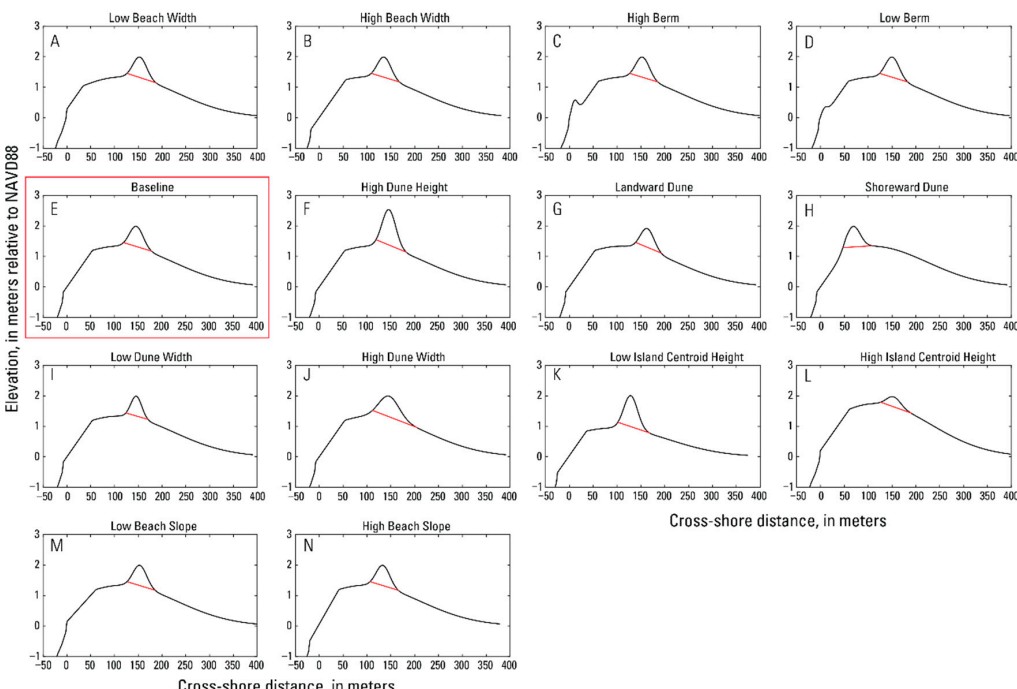

**Figure 4.** Idealized initial profiles varying island characteristics as described in Table 1; (**A**) low beach width case, (**B**) high beach width case, (**C**) high berm height case, (**D**) low berm height case, (**E**) baseline case, (**F**) high dune height case, (**G**) landward positioned dune case, (**H**) shoreward positioned dune case, (**I**) low dune width case, (**J**) high dune width case, (**K**) low island centroid height case, (**L**) high island centroid height case, (**M**) low beach slope case, and (**N**) high beach slope case. Red line indicates dune footprint area examined in post-storm analyses for dune volume and width changes (see Supplementary Materials).

### 2.3. Synthetic Storms for Simulation

Each idealized profile was subjected to six synthetic storm scenarios derived from previous modeling efforts by [21,34], (four from the 2017 study and two from the 2020 study). Each of the six synthetic storms was selected to provide a spread in maximum total water levels (TWL; defined as the combination of wave run-up, tide, and surge water levels relative to NAVD88) and duration of TWL exceeding the height of the foredune toe. The TWL for each synthetic storm was calculated via the parameterization of [5], using the mean fore-shore beach slope from Table 1 along with significant wave height ($H_s$), peak wave period ($T_P$), and tide plus surge from the various synthetic storm hydrodynamic time series. The TWL threshold to determine storm duration was set as the exceedance of the average dune toe, 1.47 m (NAVD88), calculated from the LiDAR data set discussed above based on methods from [34]. Four of the synthetic hydrodynamic conditions (open circles in Figure 5A) were derived from historical storm events (1996–2006) that were discretized into 9 bins based on maximum TWL elevation and duration of exceedance above an average dune toe threshold [34]. The wave and tide conditions for all discretized storm events in each of the 9 bins were averaged together to provide a representative scenario of hydrodynamic conditions for each bin. Details for the hydrodynamic conditions of the 9 representative scenarios (wave parameters and water levels) can be found in [34]. Of the 9 representative scenarios developed, 4 were used in this modeling study due to their spread in maximum TWL and duration: storms 1, 2, 3, and 6 (Figure 5A). Based on the spread of duration between storms 3 and 6, two other synthetic storm scenarios (tropical cyclones) were provided to fill this gap since none existed from [34]. The two additional synthetic storms (storms 4 and 5; Figure 5A) were from a suite of 295 synthetic storms from the Federal Emergency Management Agency (FEMA) Risk Mapping Region IV probabilistic coastal storm surge modeling effort for the Alabama and Florida Panhandle coasts [35]. These two storms ranged between the maximum TWL and duration of scenarios

3 and 6 (Figure 5A). It must be noted that the original surge timeseries for the FEMA derived storms did not include astronomical tide. Therefore, a morphological tide sequence, which represents the average tidal conditions based on data from the NOAA NDBC tide gauge at Dauphin Island (8735180), was added to the timeseries so that maximum surge occurred during a high tide phase. Description of the morphological tide sequence applied can be found in [21]. Figure 5A illustrates the relationship between maximum TWL and duration above the dune toe threshold for the six synthetic storms used in this study. Timeseries plots for each synthetic storm are shown in Figure 5B (aligned at the time of maximum TWL) to illustrate how the magnitude and duration of TWL above the 1.47 m dune toe threshold varies between storms.

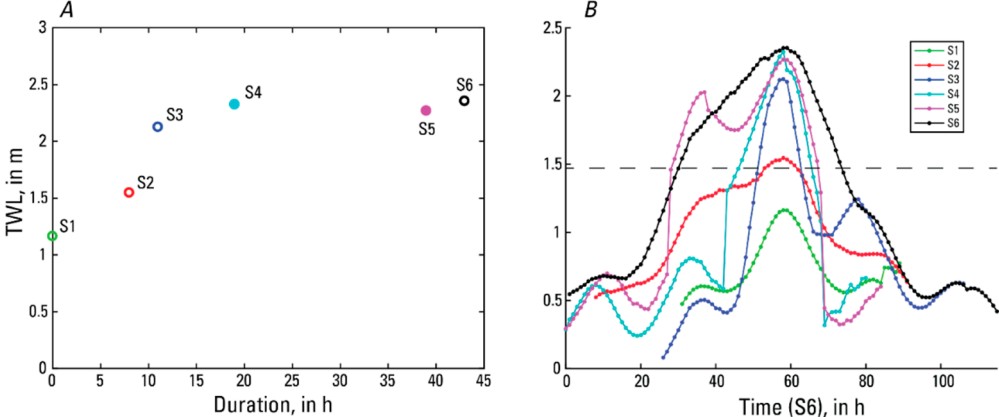

**Figure 5.** (**A**) Scatter of Total Water Level (TWL) versus duration (hours of TWL above dune toe threshold) for the 6 synthetic storms (S1–S6; open circles represent scenarios from [34], and closed circles from [21]). (**B**) TWL time series of each synthetic storm scenario aligned so that peak TWL coincides with S6 peak TWL; dashed line indicates the 1.47 m dune toe elevation threshold.

Storm 1 (S1), represented a short-lived, low magnitude (maximum TWL = 1.16 m) storm event that did not reach above the dune toe threshold; storm 2 (S2) had slightly higher maximum TWL (1.54 m) and reached above the dune toe threshold for approximately 8 h; storm 3 (S3) had a similar duration (11 h) to S2 with a higher maximum TWL (2.12 m); storm 4 (S4) had a similar maximum (2.32 m) to S3 but with a longer duration above the threshold (19 h); storm 5 (S5), was similar to S3 and S4 in maximum TWL (2.26 m) but had a much longer duration of almost 40 h; and storm 6 (S6) represented another prolonged (43 h), high magnitude (maximum TWL = 2.35 m) storm event. All storms were simulated on each idealized profile using the significant wave height ($H_s$), peak wave period ($T_p$), and tide plus surge inputs described in [34] (S1, S2, S3, S6) and [21] (S4, S5). The wave direction for each scenario was fixed to be shore-normal (180 degrees). The predicted storm impact regime based on [2], was determined for each profile for the six storm scenarios using the morphologic threshold of pre-storm dune crest and dune toe for each profile (Table 2) and is presented as a benchmark by which to compare model result variability.

**Table 2.** Storm impact regime predicted for each idealized profile and storm scenario based on [2]. X marked cells for the low dune height case which was not modeled.

| | S = Swash | | | C = Collison | | | O = Overwash | | | I = Inundation | | | | | | | | |
|---|---|---|---|---|---|---|---|---|---|---|---|---|---|---|---|---|---|---|
| | **Sallenger Regime Classifications** | | | | | | | | | | | | | | | | | |
| | **S1** | | | **S2** | | | **S3** | | | **S4** | | | **S5** | | | **S6** | | |
| | Low | Baseline | High | Low | Baseline | High | Low | Baseline | High | Low | Baseline | High | Low | Baseline | High | Low | Baseline | High |
| Beach Width | S | | S | S | | S | C | | O | O | | O | O | | O | O | | O |
| Beach Slope | S | | S | S | | S | C | | O | O | | O | O | | O | O | | O |
| Dune Height | ✕ | | S | ✕ | | S | ✕ | | C | ✕ | | C | ✕ | | C | ✕ | | C |
| Dune Width | S | S | S | S | S | S | C | C | O | O | O | O | O | O | O | O | O | O |
| Dune Position | S | | S | C | | S | O | | O | O | | O | O | | O | O | | O |
| Island Height | S | | S | C | | S | C | | O | O | | O | O | | O | O | | O |
| Berm present | S | | S | S | | S | C | | C | O | | O | O | | O | O | | O |

## 2.4. Numerical Model Setup: 1D and 2D XBeach

XBeach is a 2D process-based numerical model that solves horizontal equations for wave propagation, flow, sediment transport and bottom changes under varying spectral wave and flow boundary conditions and was developed to study nearshore dune and barrier island response to storm events [6]. Wave group scale equations for roller energy, time-dependent short-wave action balance, nonlinear shallow water mass and momentum, and sediment transport are concurrently solved in XBeach ([6]). XBeach can be set up in a 1D parameter space [17,36–38] or a 2D parameter space [22–25,39] depending on the appropriateness of the application, computational efficiency, and research goals. The 2D setup of the XBeach model has been validated at Dauphin Island by [24] for morphological changes as a result of Hurricanes Ivan and Katrina. Both 1D and 2D setups were utilized in this study using the same model parameters for comparison and evaluation for future study.

The 2D simulations were setup by replicating the 1D idealized profiles in the alongshore with a resolution of 25 m resulting in a rectangular grid 30 km alongshore and approximately 6.5 km cross-shore (Figure 3D). The 2D model setup was used to better approximate the response of the profile considering both alongshore and cross-shore hydrodynamics and sediment transport dynamics [38]. XBeach model parameters for all 2D simulations were setup like those in [34], with the 'single_dir' option in place and a morphological acceleration factor of 10, with all other parameters set to default. For the 2D simulations, a buffer area extending approximately 2.5 km from the lateral boundaries of the model domain was excluded from post-storm topographic analysis to eliminate any wave boundary effects that might impact the profiles (Figure 3D). The 1D setup simulations used the same model parameters, however, preliminary 1D results showed greater erosion compared to the 2D setup. To replicate reasonable corresponding morphological changes from the 2D setup in a 1D setup, sensitivity tests that varied the model bed friction coefficient (BFC) parameter were performed which found that a spatially constant BFC of 44 was better suited than the default BFC parameter of 55.

## 2.5. Profile Change Metrics

By comparing the variability in profile response between the baseline and modified profiles, the degree to which antecedent morphologic features influence storm-induced morphological change can be identified. Analyses included calculating changes to beach width, beach slope, beach volume, maximum island elevation (dune and island base), island width, shoreline retreat distance, dune width, and dune volume. Post-storm beach characteristics were based on the post-storm derived fore-dune toe and shoreline location. Changes to dune volume and width were calculated based on the changes to the profile within the initial dune footprint, with dune volume ($D_v$) calculated as the area under the dune portion of the profile (area above red line in Figure 4). Refitting a Gaussian curve to the post-storm profile can introduce fitting errors unrelated to profile response, therefore pre- and post-storm fore-dune width ($D_w$) was identified as the width of the initial dune footprint pre-storm and the width of the remaining dune feature within the initial dune footprint post-storm; if no portion of the profile was present within the initial dune footprint post-storm $D_w$ was 0. The volume of overwash deposits was calculated as the amount of sediment above the pre-storm profile landward of the dune footprint position. For the 2D simulations, post-storm island characteristics were determined for each alongshore profile within the model domain but excluded the 2.5 km buffer areas on the lateral boundaries. The mean and standard deviation across the analysis area were calculated and compared to the pre-storm profile characteristics for each idealized profile. For the 1D simulations, the post-storm characteristics were derived from the post-storm profile directly for comparison.

## 3. Results

### 3.1. Morphologic Response

The pre- and post-storm profiles subjected to the varying storm conditions are shown in Figure 6 (1D) and Figure 7 (2D). The quantitative analysis of morphologic change for each idealized profile under

the varying storm conditions for both dimensional setups are shown in Figures 8–11, presented as percent change relative to the pre-storm values. These qualitative and quantitative results were compared to one another for the corresponding storm scenarios and the predicted storm impact regime (Table 2). Additionally, modified idealized profile response was compared to the baseline idealized profile response for each storm scenario to illustrate and identify uncertainty related to antecedent morphology.

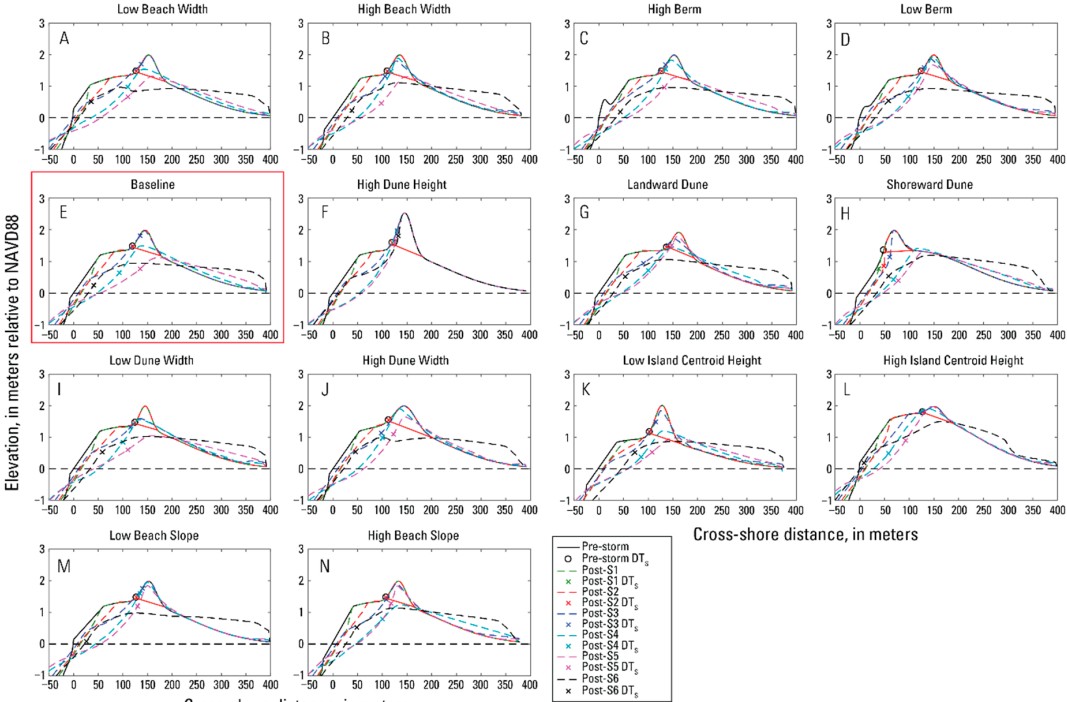

**Figure 6.** 1D pre- and post-simulation profiles for all 6 storm scenarios with shoreward dune toe position (DT$_s$) identified. (**A**) low beach width case, (**B**) high beach width case, (**C**) high berm height case, (**D**) low berm height case, (**E**) baseline case, (**F**) high dune height case, (**G**) landward positioned dune case, (**H**) shoreward positioned dune case, (**I**) low dune width case, (**J**) high dune width case, (**K**) low island centroid height case, (**L**) high island centroid height case, (**M**) low beach slope case, and (**N**) high beach slope case. Horizontal dashed line denotes the 0 m elevation line for reference. Red line indicates dune footprint area examined in post-storm analyses for dune volume and width changes.

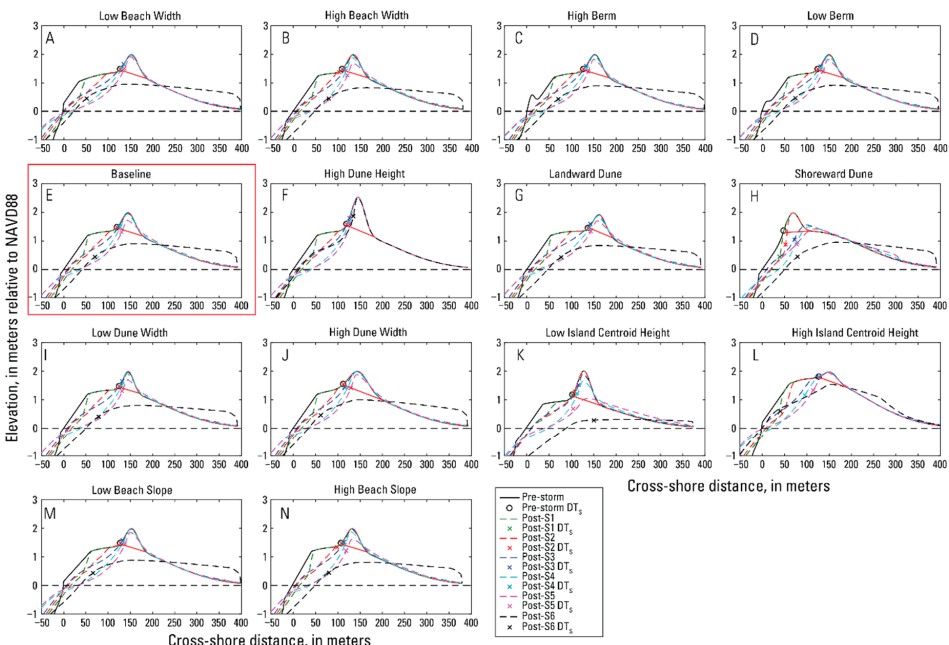

**Figure 7.** 2D pre- and averaged post-simulation profiles for all 6 storm scenarios with alongshore averaged shoreward dune toe position (DT$_s$) identified. (**A**) low beach width case, (**B**) high beach width case, (**C**) high berm height case, (**D**) low berm height case, (**E**) baseline case, (**F**) high dune height case, (**G**) landward positioned dune case, (**H**) shoreward positioned dune case, (**I**) low dune width case, (**J**) high dune width case, (**K**) low island centroid height case, (**L**) high island centroid height case, (**M**) low beach slope case, and (**N**) high beach slope case. Horizontal dashed line denotes the 0 m elevation line for reference. Red line indicates dune footprint area examined in post-storm analyses for dune volume and width changes.

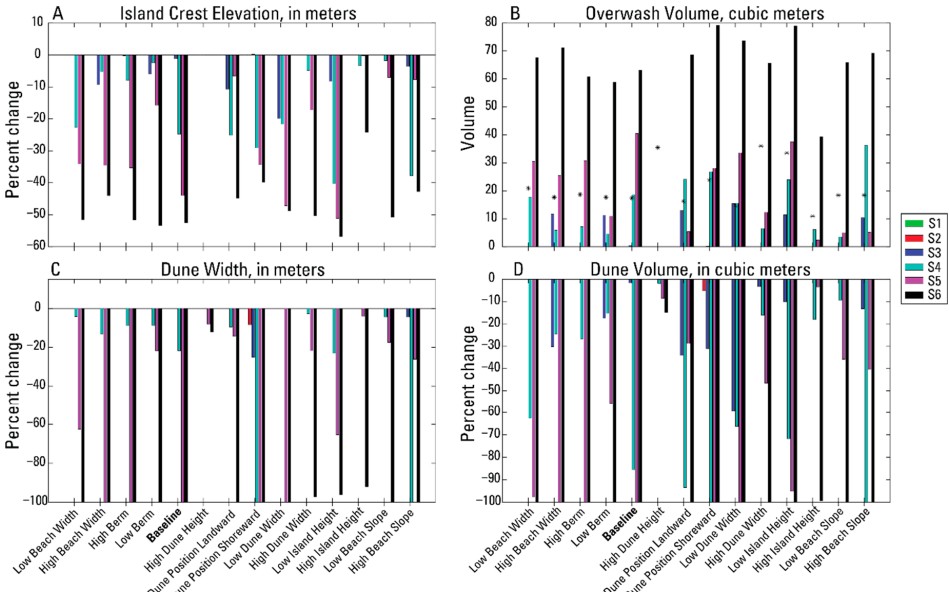

**Figure 8.** 1D post-storm percent change to the maximum island crest elevation (**A**), dune width (**C**), and dune volume (**D**) for storms 1 through 6 (S1–S6). 1D post-storm calculated overwash volume (**B**); note that asterisks indicate the pre-storm dune volume. Metrics are provided in the panel titles to indicate those features that were analyzed, not the values presented in the graphs. Note: *y*-axis scales differ among panels.

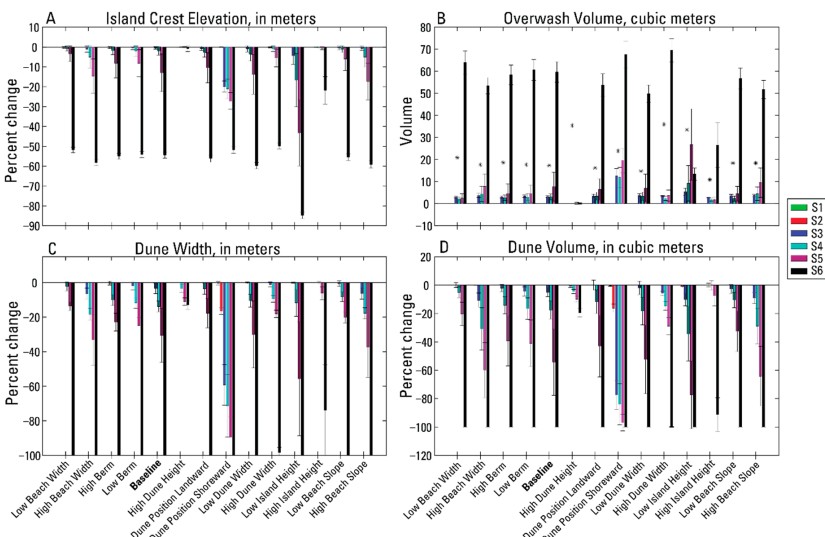

**Figure 9.** 2D post-storm percent changes to the alongshore averaged (with error bars of standard deviation) maximum island crest elevation (**A**), dune width (**C**), and dune volume (**D**) for storms 1 through 6 (S1–S6). 1D post-storm calculated overwash volume (**B**); note that asterisks indicate the pre-storm dune volume. Metrics are provided in the panel titles to indicate those features that were analyzed, not the values presented in the graphs. Note: *y*-axis scales differ among panels.

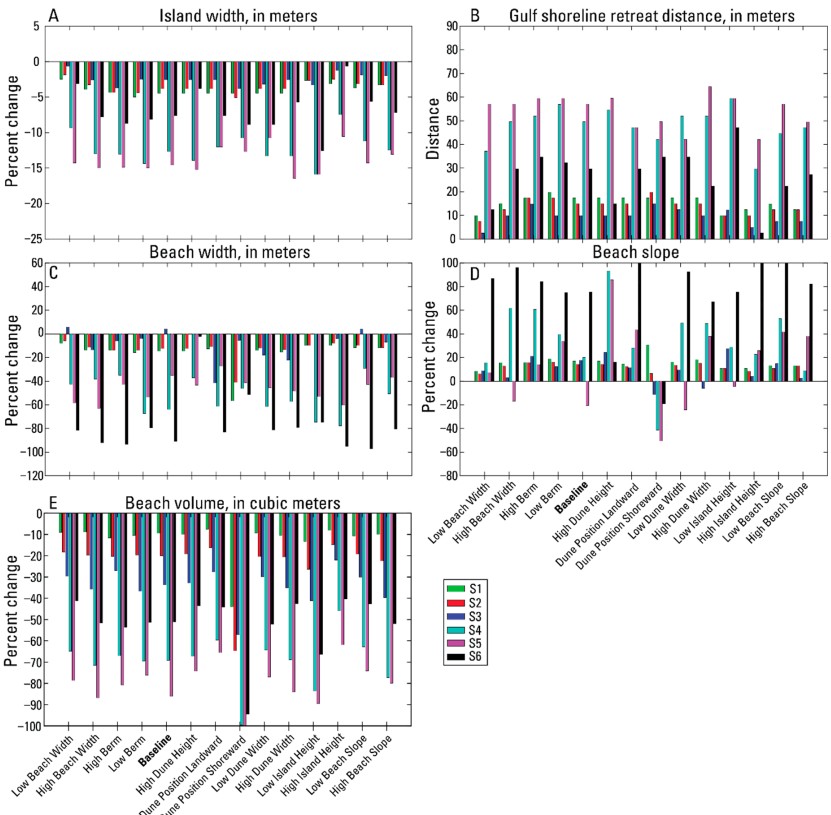

**Figure 10.** 1D post-storm percent change to island width (**A**), beach width (**C**), beach slope (**D**), and beach volume (**E**) for storms 1 through 6 (S1–S6). 1D post-storm calculated gulf shoreline retreat distance (**B**). Metrics are provided in the panel titles to indicate those features that were analyzed, not the values presented in the graphs. Note: *y*-axis scales differ among panels.

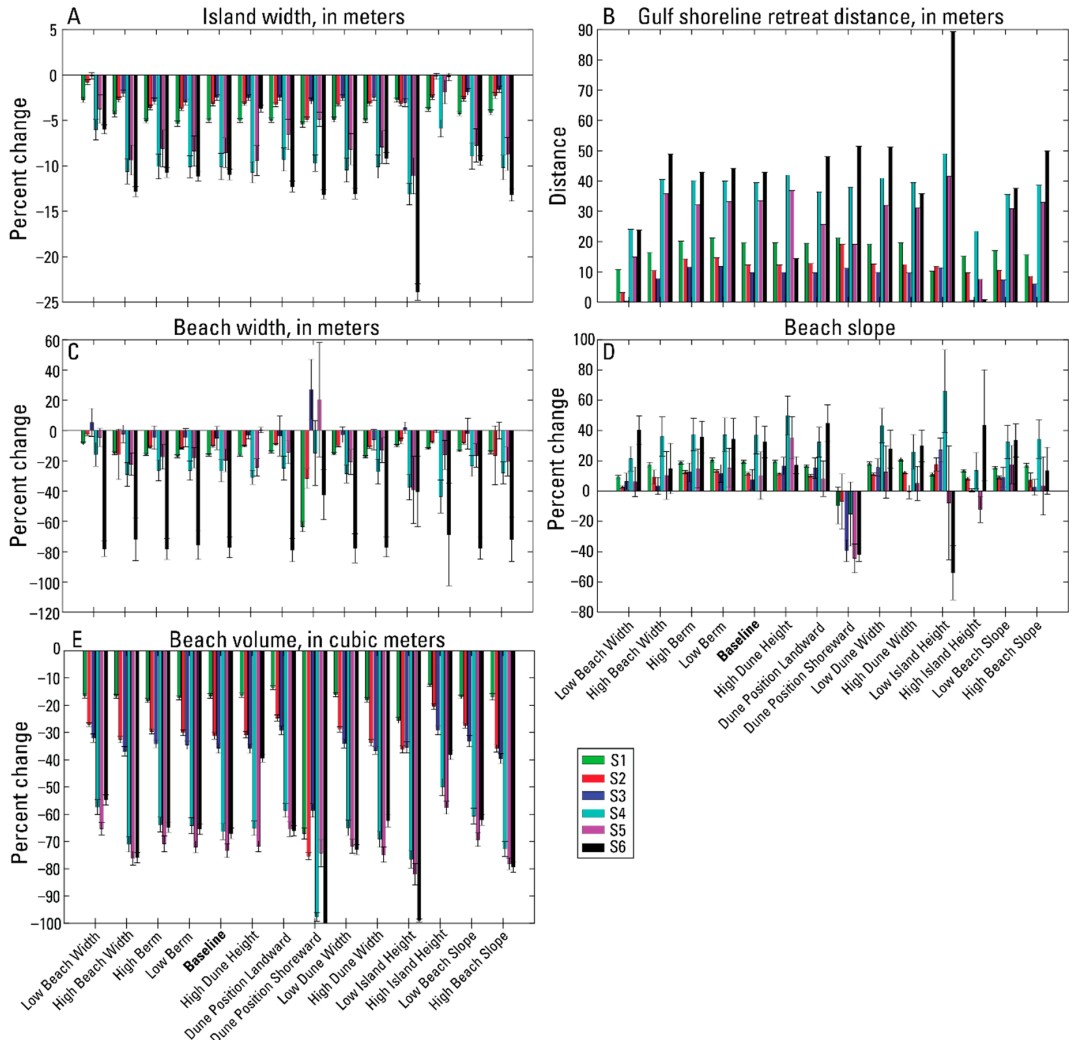

**Figure 11.** 2D post-storm percent changes to the alongshore averaged (with error bars of standard deviation) island width (**A**), beach width (**C**), beach slope (**D**), and beach volume (**E**) for storms 1 through 6 (S1–S6). 2D post-storm alongshore averaged gulf shoreline retreat distance (**B**). Metrics are provided in the panel titles to indicate those features that were analyzed, not the values presented in the graphs. Note: *y*-axis scales differ among panels.

The swash regime, in which no wave-dune interactions occur, was predicted for S1 and S2 in most of the profile simulations, except for collision predicted for the shoreward positioned dune and low island height profiles (Table 2). For these two storm scenarios, the beach area was more impacted than the dune feature in both dimensional setups of all profile cases except for the shoreward positioned dune profile (Figures 10 and 11). This profile has a small beach area and a high slope transition between beach and dune features so that even with low water levels dune avalanching can result through erosion of the upper beach and undermining of the dune face.

As storm scenario magnitude and duration increased, the 1D and 2D results began to diverge compared to their predicted impact regime (Table 2). The results for S3 through S5 show that there is a clear dimensional setup difference in morphologic response. The morphologic response of the dune for S3 in the 1D setup was slightly greater than that observed for the 2D setup in a few cases. The landward migration of the dune-toe for S3 shows that dune collision was minor but prevalent among the varying simulations (Figures 6 and 7), which follows the predicted impact regimes. For the profiles predicted to experience overwash during S3 (Table 2), none was observed in the 1D setup and only the shoreward positioned dune profile experienced overwash in the 2D setup. For S4 and S5,

the profile and morphological change for the 1D setups were much greater in most of the simulations compared to the 2D setup. The S4 simulations in 1D all had some dune volume loss either through dune avalanching or overwash, the former being more likely for cases when overwash volumes were less than 10 m$^3$. Conversely, the 2D simulation response for S4 was more indicative of the collision regime based on the low dune volume and island crest elevation changes (Figure 9A,D). A similar pattern of profile response was observed for S5 although to a greater magnitude in both setups with more instances of overwash than in S4. Both dimensional setups for S6 were observed to have high instances of overwash based on morphological response and post-storm profile configuration (Figures 6 and 7) except for the high dune height profile.

### 3.2. Baseline Deviations and Uncertainty

The relative difference between baseline and modified profile response of various characteristics was analyzed by comparing the ratio of percent change to profile characteristics of all modified profiles to that of the baseline. These ratios are shown in Figure 12 (1D) and Figure 13 (2D) on a logarithmic scale to visualize the magnitude of differences (values equal to 0 were set to 10$^{-5}$ for inclusion in Figures 12 and 13). In some cases, one of the relative difference values was less than 0. In Figures 12 and 13, a black square indicates that the pre- to post-storm difference in the baseline simulation was 0 or positive while the modified profile difference was negative. A black circle indicates the pre- to post-storm difference in the baseline simulation was negative while the modified profile difference was positive. These represent cases where modifications to profile characteristics lead to an opposite effect compared to the baseline profile. For dune volume (DV), island crest elevation (IC), and overwash volume (OV), there were no changes in S1 in either dimensional setup. For S2, there were only changes to the DV for the shoreward positioned dune profile in both setups (note the negative values here are based on no change to baseline profile DV). For DV, there was marked variation in the differences in response of modified profiles compared to the baseline, specifically for the 1D setup in S3, S4, and S5 (Figure 12). The variation in the 1D response difference for S3 of the modified profiles was either 0 or ranged from 1 to 40 times the dune volume change of the baseline profile, while S4–S5 had magnitude differences in some cases slightly greater than 1 but more generally less than 1, indicating the change to the baseline profile was greater than changes to the modified profile for these storm scenarios. A less defined pattern was observed for the 2D setup (Figure 13) where the response difference of S3 was not as pronounced and S4–S5 responses shifted closer to or greater than 1 in more cases compared to the 1D setup. The IC response differences followed a similar pattern as DV with S3 having the largest range of differences in the 1D setup (Figure 12C); the 2D setup showed minimal changes of IC in S3 for a few modified profiles (note the negative values here are based on no change to baseline profile IC), but S4 and S5 had a shift of response differences to greater than 1 (Figure 13C) in more profiles compared to the 1D setup of these simulations. The 1D OV differences, again, had a much higher magnitude range for S3 compared to S4–S5 (Figure 12D), although for the 2D setup the ranges between these three storms were similar and smaller with a shift to a difference in magnitude closer to or greater than 1 (Figure 13D). The response difference to the dune width (DW) was the most diverse among the storm scenarios for dune characteristics, in that no single storm scenario had a patterned response difference compared to others aside from S6 which was close to 1 for most other characteristics as well. S4 and S5 had the greatest difference in response compared to the baseline unlike the other three characteristics previously described for the 1D cases, whereas S3 had the highest range in response differences for the 2D setup (note the negative values in both 1D and 2D are based on no change to baseline profile DW). A total of 18 profile scenarios were evaluated showing change in the dune characteristics, including OV, for the 1D setup and a total of 19 profile scenarios were evaluated showing change in the 2D setup; cases with no difference in baseline and modified profile dune changes were excluded. The profiles with modified dune position and island base height had larger changes to these characteristics compared to the baseline among all the modified profiles evaluated. The shoreward positioned dune profile had the greatest amount of change in 6 of the 18

evaluated cases in 1D (S2: DV, DW; S3: DW, S4: DV, DW; S6: OV) and 12 of the 19 evaluated cases in 2D (S2: DV, DW; S3 and S4: DV, DW, IC, and OV; S5: DV, DW), while the low island height profile had the second highest amount of change in 4 of the 18 evaluated cases in 1D (S4: IC; S5: IC, OV; S6: IC) and 3 of the 19 evaluated cases in 2D (S5: IC, OV; S6: IC). Conversely, the high dune height profile had the lowest amount of change (0 which was set to $10^{-5}$ on the logarithmic *y*-axis) compared to the baseline profile for 8 of the 18 cases evaluated in the 1D setup (S4: DV, IC, OV; S5: OV; S6: DV, DW, IC, OV), while the high island height had the lowest amount of change for 2 of the 18 cases evaluated (S5 DV, DW). It is important to note, however, that for the remaining 8 cases, the response difference of 0 was the lowest observed for both these modified profiles. For the 2D setup, this pattern continued but with the high dune height cases having the lowest amount of change compared to the baseline for 6 of the 19 cases evaluated (S3: OV; S4: OV; S5: OV; S6: DV, DW, IC, OV), and 4 cases for the high island height profile (S4: DV, DW; S5: DV, DW), again for the remaining 8 cases, the response difference of 0 was the lowest observed, which was the case for both these modified profiles.

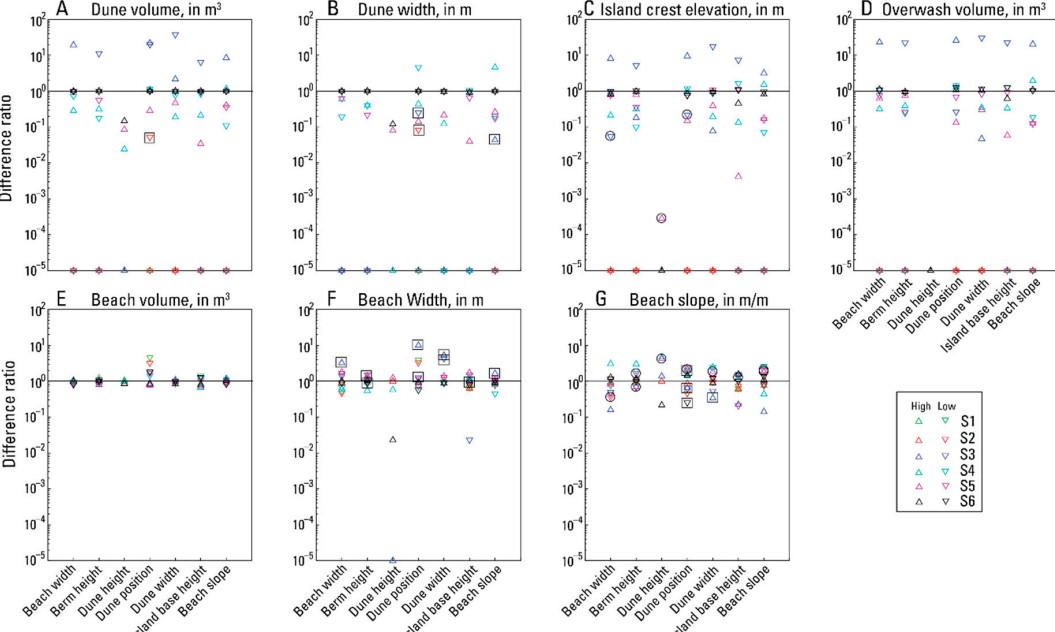

**Figure 12.** 1D difference ratio of percent change of dune volume (**A**), dune width (**B**), island crest elevation (**C**), overwash volume (**D**), beach volume (**E**), beach width (**F**), and beach slope (**G**) of the baseline and modified profiles; note circled markers around storms markers (S1–S6) indicate a negative relative response difference. Black squares indicate the pre- to post-storm difference in the baseline simulation was 0 or positive and black circles indicate the pre- to post-storm difference in the baseline simulation was negative. Horizontal black line indicates a difference ratio of 1; undefined or 0 ratio values were set to $10^{-5}$ for inclusion in the figures.

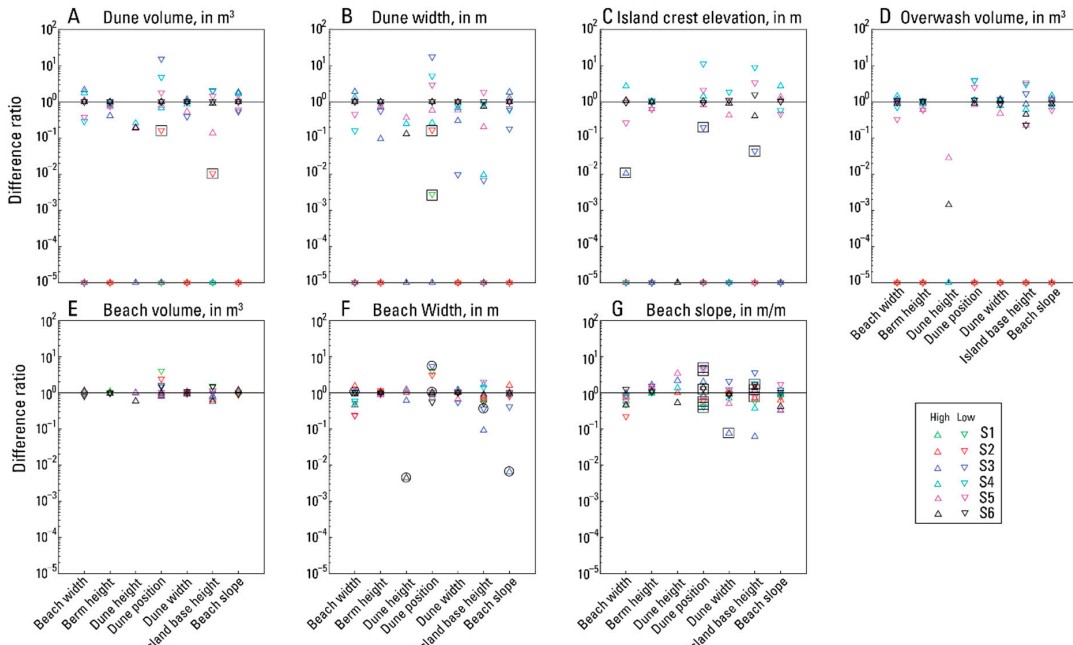

**Figure 13.** 2D difference ratio of percent change of alongshore averaged dune volume (**A**), dune width (**B**), island crest elevation (**C**), overwash volume (**D**), beach volume (**E**), beach width (**F**), and beach slope (**G**) of the baseline and modified profiles; note circled markers around storm markers (S1–S6) indicate a negative relative response difference. Black squares indicate the pre- to post-storm difference in the baseline simulation was 0 or positive and black circles indicate the pre- to post-storm difference in the baseline simulation was negative. Horizontal black line indicates a difference ratio of 1; undefined or 0 ratio values were set to $10^{-5}$ for inclusion in the figures.

The response difference of beach characteristics was less variable than those illustrated for dune characteristics. Response differences in beach volume (BV) were minor for all scenarios in both the 1D and 2D setups, except for the shoreward positioned dune profile which ranged up to 4.5 times the baseline for the 1D, and 4 times the baseline for the 2D (Figures 12E and 13E). For beach width (BW), S3 again had the highest range and magnitude of difference in both the 1D and 2D setups (Figures 12F and 13F). Note that S3 had difference magnitudes that ranged from −10 to less than 1 times the baseline for the 1D and −6 to less than 1 times the baseline for the 2D, while other scenarios had ranges slightly less than this for both dimensional setups. Interestingly, the negative values in the 1D and 2D BW comparisons are opposite, with baseline changes being positive in the 1D and negative in the 2D. This, again, suggested that the dimensional setup has some influence on the evolution of the varied profile response. The beach slope (BSL) response difference among scenarios was highly variable with no discernable pattern, S4 had the highest range of values for the 1D cases (4.6), while S3 had a larger range of response difference for the 2D cases (−5.2; Figures 12G and 13G). A total of 18 profile scenarios were evaluated showing change in the beach characteristics for the 1D and 2D setups; cases with no difference in baseline and modified profile beach changes were excluded. The shoreward positioned dune profile dominated the maximum response difference for the beach characteristic with 10 of the 18 evaluated cases in 1D (S1: BV, BW, BSL; S2: BV, BW; S3: BV, BW; S4: BV; S5: BV; S6: BV) and 10 of the 18 evaluated cases in 2D (S1 and S2: BV, BW; S3: BV, BW, BSL; S4: BV; S5: BSL; S6: BV). The high base island height profile was observed to have the least amount of change compared to the baseline for both dimensional setups, with 5 of the 18 cases in 1D (S2–S6: BV) and 8 of the 18 cases in 2D (S1: BV; S2: BV; S3 and S4: BV, BSL; S5 and S6: BV).

## 4. Discussion

The results of these simulations illustrate how different antecedent profile characteristics and their varying morphologic thresholds can influence the evolution of morphological characteristics, the effect of which scales with storm magnitude and duration for most cases. These relationships are easily identified in the 1D and 2D setups even though there was a marked difference in the amount of erosion in the medium level storm scenarios between these two setups (Figures 8 and 9). Interestingly, the similar response and magnitude of erosion for the beach and dune features for almost all the modified profiles for S1 and S2 in both dimensional setups suggest that XBeach tends to evolve different initial barrier island configurations toward a similar post-storm shape for less energetic storms. Additionally, like low magnitude storm conditions, XBeach simulations of the highest magnitude storm condition (S6) tended to evolve varying initial barrier island profiles to a similar post-storm shape. As for the medium level storm scenarios (S3–S5), dimensional setup differences have been shown in other studies, as well as the sensitivity testing, where the 1D setup of XBeach simulations tend to overpredict wave run up compared to a 2D setup of the same conditions which can directly correlate to beach and dune erosion [37,40]. This is evident by the response to the medium level storm scenarios (S3–S5) in the 2D case which were almost exclusively characterized by dune collision (Figure 7) while in the 1D setup the response was more varied with instances of overwash and extensive dune collapse (Figure 6). These differences in the mid-level storm magnitudes illustrate where non-static morphological thresholds, such as dune crest elevation decreases due to collapse or overwash, become very important. Accounting for the time-varying morphologic change, such as dune avalanching, throughout a simulation is critical to predicting regimes beyond using water level predictions alone. Through extensive modeling efforts with a larger range of storm magnitudes and durations, as well as expansion of the morphological dataset, the quantification of time-varying morphologic change can be captured and explored further for inclusion in coastal change hazard predictions.

For all modified profiles, their comparison to the baseline case for each scenario has shown that uncertainty in profile evolution can be attributed to antecedent morphology, morphologic evolution throughout a storm event, storm magnitude, or a combination of these factors. For dune features, the comparisons shown in Section 3.2 illustrate that while variation exists for all scenarios, the storm scenarios that represent medium magnitude storms (S3–S5) exhibit the greatest differences and therefore lead to more uncertainty. Specifically, the 1D response differences compared to the baseline for DV, IC, and OV in S3 (Figure 12) show that storm magnitude can play a major role in the variation in response for specific cases. The response difference for most of these cases when compared to the baseline is high even among highly varied initial profile morphologies. However, the lack of this pattern in the 2D setup suggests that the results from the 1D setup of the S3 cases could be highly dependent on the dimensional response difference instead of the storm magnitude. Putting dimensional setup aside and focusing on the response deviation of the modified profiles from the baseline shows that varied dune position and dune width are consistently highest for S3 in the 2D and 1D setup, respectively. The shoreward positioned dune has the greatest relative change for DV, DW, IC, and OV compared to the baseline in the 2D setup, while in the 1D setup the low dune width profile had the greatest relative change for DV, IC, and OV. For the latter case, the time-varying evolution of the antecedent morphology through the storm event causes beach slope to increase thereby increasing wave run-up and erosion of the dune feature. The high response difference for the shoreward positioned dune is directly related to the initial profile setup more so than the time varying evolved profile. In this case, the initial morphology of the dune feature was closer to the shoreline which caused earlier interaction between waves and the dune leading to more eroded features compared to the baseline.

For S4 and S5, the varied response in both dimensional setups is more attributable to the antecedent morphology and the time-varying evolution of that morphology throughout the storm scenarios than to the storm magnitude. Specifically, for the 1D cases there was a tendency for response differences in dune characteristics of the modified profiles to be less than or equal to the baseline magnitude (Figure 12). For the modified profiles that had response differences less than 1 in S4, in most cases the time-varying

beach slope was greater at the peak of the storm for the baseline profile which likely led to greater wave run-up and dune erosion compared to the modified profiles. Alternatively, in a few of those cases, specifically the high dune height, high dune width, and high island base height, the antecedent morphology played a larger role in reducing the amount of erosion that occurred. These three profiles increased the pre-storm morphologic thresholds of dune crest elevation, dune volume, or dune toe elevation, respectively, which reduced the vulnerability of the dune feature to wave impacts compared to the baseline profile. For S5, the antecedent morphology played a much larger role in reducing the response difference compared to the baseline. Along with the three profiles just mentioned, the initial morphology of the landward positioned dune profile reduced the wave-dune interactions by essentially increasing the beach width and therefore lowering the slope for wave run-up. The other modified profiles that had response differences less than 1, followed the pattern as in S4 with time-varying slope reduction at or before the peak of the storm which likely led to reduced wave run-up compared to the baseline profile. For the 2D simulations, there was less distinction in the difference response for most of the dune characteristics evaluated, however for S4 and S5 there was an observable shift in differences toward the baseline change magnitude. This could be related to the overall difference in the dimensional setups for these two storm scenarios which had a reduction in the amount of erosion of the dune feature (Figures 8 and 9), as well as a reduction in the variability in erosion among the modified profiles.

The comparisons of beach characteristic changes between the baseline and modified profiles are less varied than those associated with dune characteristics. Specifically, the response difference in BV was minimal for almost all modified profiles and all scenarios in both the 1D and 2D setups, the main exceptions being the shoreward positioned dune and low island height profiles. The initial morphology of these two modified profiles is likely the root for these differences due to the volume loss of beach sediment of the initial morphology compared to the baseline. While these analyses compare relative differences based on normalized volumes, the reduction in overall beach volume of these two profiles led to high losses when impacted by the varying storm conditions (Figures 10 and 11). Additionally, for both dimensional setups the scenarios with the greatest differences in BV loss were S1 and S2 which suggests that lower magnitude storms can have a larger effect on the beach area compared to higher magnitude storms that exhibit wave-dune interactions that can transport sediment from the dune to the beach. The changes and response differences for BW and BSL give no clear indication of whether antecedent morphology or storm magnitude play a larger role, however for BW in the 1D simulations there is an observable pattern of S4 having response differences less than 1 in most cases while S5 response differences are mostly greater than 1. This is likely due to the difference in the duration of elevated TWL levels (Figure 5), where S5 had an earlier ramp up of storm conditions which likely increased the beach erosion in most cases compared to the later ramp up time and shorter duration of S4. Conversely, for the 2D setup the response differences were highly variable with increased uncertainty as to which factor plays a more significant role in profile evolution.

The time-varying evolution of beach slope is highly important when accounting for the post-storm changes between modified profiles and storm magnitudes. For example, in two cases for the 1D setup (landward dune and high beach slope), S4 resulted in greater dune erosion than S5 even though S5 has a much longer duration of elevated TWL. Investigation of morphologic evolution through time for these two simulations shows that the long period of elevated TWL before the time of peak TWL during S5 resulted in a shallow beach slope forming before peak TWL, and therefore low wave runup at this time. Conversely, the shorter duration of elevated TWL for S4 did not allow a shallow beach slope to form which caused higher wave run-up comparatively at the time of peak TWL and thus more dune erosion for both the landward dune and high beach slope simulations (Figure 6G,N). This reinforces the importance of accounting for time-varying morphologic characteristics, specifically beach slope, and their effect on predicting barrier island evolution.

## 5. Conclusions and Future Work

The suite of idealized profiles, synthetic storms, and morphologic responses presented in this paper has provided insight into how the process-based model XBeach evolves different barrier island profiles under the same hydrodynamic conditions. This study has shown that morphologic evolution in XBeach is sensitive to morphologic thresholds related to elevation and slope, both static and time-varying, especially for the medium storm magnitudes outlined in this study. Additionally, the setup of XBeach simulations in 1D has been observed to have larger morphologic changes to most of the profile setups compared to changes from the 2D simulations. Conversely, the model results indicate that XBeach simulation of low and high magnitude storm conditions usually evolve idealized profiles of a low elevation barrier island to a similar post-storm shape regardless of the pre-storm configuration or dimensional setup. However, there is a need for more sensitivity testing as it relates to determining what model parameters need modification in order to reproduce comparable results of 1D and 2D simulations, which are necessary for increasing the computational efficiency for a larger operational framework. The sensitivity to beach slope, whether modified by manipulating antecedent morphology or time-varying throughout a storm event, and its effect on morphologic change was highlighted in a way that magnifies its importance for coastal scientists studying or predicting barrier island morphologic response to storm impacts. This study has shown how varied antecedent morphology can influence differences in storm impact regardless of dimensional setup in XBeach with increases to dune volume (i.e., height or width) or island base height decreasing vulnerability to dune erosion, whereas dunes positioned closer to the shore (or by proxy a narrow beach width) and a reduction in island base height increased dune vulnerability. Building upon and expanding the scope of this work by incorporating larger and multiple dune features, wider beaches, vegetated dunes, nearshore sandbars, presence of a tidal bay or marsh flats, wider range of hydrodynamic conditions, and greater sensitivity analysis of dimensional setup differences could broaden the understanding of storm induced morphological response in XBeach and its use in an operational framework for predicting coastal change hazards.

Future work to expand the range of morphological characteristics, derived from multiple sites, could provide a greater variety of coastal environment representation. In addition, constraining observed data sets to specific island states that represent pre-storm, post-storm, and recovered states in order to provide morphological characteristic values for each state could enhance the representativeness of the idealized profiles to these specific island states and account for the correlation expected between individual profile characteristics. This would inform determination of coastal resiliency, restoration, and identification of vulnerabilities by providing specific profile configurations for the island states and determining how each respond to varying storm impacts. By applying these methods to different and more diverse coastal environments, a database of morphological characteristics, along with their morphological response to various storms could be harnessed to build on the work of others [15–17] to form an operational model for predicting magnitudes of coastal change, as well as a research tool for understanding how diverse coastal systems react to coastal change hazards.

**Supplementary Materials:** The following are available online at https://doi.org/10.5066/P9VD60JC, 1-dimensional coordinates (UTM Easting and UTM Northing) and topographic/bathymetric elevations for all initial profiles in Figure 4.

**Author Contributions:** Conceptualization, R.C.M. and P.S.D.; methodology, R.C.M. and P.S.D.; formal analysis, R.C.M., P.S.D., and R.M.; investigation, R.C.M.; writing—Original draft preparation, R.C.M.; writing—Review and editing, P.S.D., R.M., and D.P.; visualization, R.C.M., P.S.D., and R.M.; supervision, P.S.D. and D.P.; project administration, P.S.D. and D.P.; funding acquisition, P.S.D. and D.P. All authors have read and agreed to the published version of the manuscript.

**Funding:** This research was funded by the U.S. Geological Survey's Coastal and Marine Hazards and Resource Program.

**Acknowledgments:** R.C.M. would like to thank Ap van Dongeren and Ellen Quataert of Deltares for early help in the review of the methodology. The authors would also like to thank Justin Birchler and the two anonymous reviewers for providing input on this study. Any use of trade, firm, or product names is for descriptive purposes only and does not imply endorsement by the U.S. Government.

**Conflicts of Interest:** The authors declare no conflict of interest.

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
