# Peer review of "Sensitivity of Storm Response to Antecedent Topography in the XBeach Model"

_jmse, doi:10.3390/jmse8100829_

Round 1

Reviewer 1 Report

This work investigates the sensitivity of storm response to antecedent topography in the XBeach Model. The object is within the scope of the JMSE journal. My suggestion is a minor revision.

  1. Line 85, “2.1. Study area nd LiDAR analysis” should be “2.1. Study area and LiDAR analysis”.
  2. Antecedent topography played a very important role in coastal evolution. Four classes of antecedent topography have been recognized (James and Macintyre, 1985) currently. Which class of antecedent topography was used in the study?
  3. Line 566 and 567, “This study has shown that morphologic evolution in XBeach is sensitive to morphologic thresholds”. What are morphologic thresholds? More details should be provided in the manuscript.

Reviewer 2 Report

Storm induced beach changes are caused by both hydrodynamic conditions and the antecedent beach profiles. Nowadays, most of our studies are focusing on the hydrodynamic part, while the effects of antecedent beach status are usually ignored. Part of the reason is the difficulties to access the large-scale pre-storm beach profiles. 

Based on our field observation along the Gulf coast, I do observed that the pre-existing beach profiles play a very important role in response to storm impacts. I feel it is challenging to address the mechanism of storm-induced beach changes induced by different antecedent beach status. I am very happy to see this manuscript as this paper directly address this challenging topic. The Xbeach model used in the paper is a very well known open source software and has a very good reputation in our field. It is based on solid dataset over long-term temporal scale, it also have a reasonable spatial coverage. The results really help us to understand the underlying mechanism of various beach changes based on different antecedent profiles.   

This paper is well written and well organized. I recommend publication of this paper in the present form. 
